# Loss of the melanocortin-4 receptor in mice causes dilated cardiomyopathy

Michael J Litt[1], G Donald Okoye[2,3], Daniel Lark[1], Isin Cakir[4], Christy Moore[5], Mary C Barber[2,3], James Atkinson[6], Josh Fessel[5], Javid Moslehi[2,3], Roger D Cone[1,4,7]*

[1]Departments of Molecular Physiology and Biophysics, Vanderbilt University, Nashville, United States; [2]Division of Cardiology, Vanderbilt University, Nashville, United States; [3]Cardio-Oncology Program, Department of Medicine, Vanderbilt University, Nashville, United States; [4]Life Sciences Institute, University of Michigan, Ann Arbor, United States; [5]Allergy Pulmonary and Critical Care, Vanderbilt University Department of Medicine, Nashville, United States; [6]Department of Pathology, Vanderbilt University Medical Center, Nashville, United States; [7]Department of Molecular and Integrative Physiology, University of Michigan School of Medicine, Ann Arbor, United States

**Abstract** Haploinsufficiency of the melanocortin-4 receptor, the most common monogenetic obesity syndrome in humans, is associated with a reduction in autonomic tone, bradycardia, and incidence of obesity-associated hypertension. Thus, it has been assumed that melanocortin obesity syndrome may be protective with respect to obesity-associated cardiovascular disease. We show here that absence of the melanocortin-4 receptor (MC4R) in mice causes dilated cardiomyopathy, characterized by reduced contractility and increased left ventricular diameter. This cardiomyopathy is independent of obesity as weight matched diet induced obese mice do not display systolic dysfunction. *Mc4r* cardiomyopathy is characterized by ultrastructural changes in mitochondrial morphology and cardiomyocyte disorganization. Remarkably, testing of myocardial tissue from *Mc4r*−/− mice exhibited increased ADP stimulated respiratory capacity. However, this increase in respiration correlates with increased reactive oxygen species production – a canonical mediator of tissue damage. Together this study identifies MC4R deletion as a novel and potentially clinically important cause of heart failure.

DOI: https://doi.org/10.7554/eLife.28118.001

*For correspondence: rcone@umich.edu

**Competing interests:** The authors declare that no competing interests exist.

## Introduction

During the last 30 years, obesity has become a leading cause of morbidity and mortality (*Ogden et al., 2014*; *Christakis and Fowler, 2007*). Health risks associated with obesity include type 2 diabetes, hypertension, and coronary artery disease. Obesity is also an independent risk factor associated with the development of heart failure (*Kenchaiah et al., 2002*) (*Levitan et al., 2009*). Haploinsufficiency of the *Mc4r* is the most common monogenetic obesity syndrome in man (*Farooqi et al., 2003*) and is responsible for 0.5–2.5% of all early onset morbid obesity (*Stutzmann et al., 2008*) making it an important consideration in personalized obesity care. The prevalence of the *Mc4r* obesity syndrome is a result of its dominant inheritance pattern, and penetrance of approximately 70% (*Tarnow et al., 2008*) (*Biebermann et al., 2003*; *Tao, 2005*).

*Mc4r* is expressed most heavily in the central nervous system where it plays a critical role in energy balance (*Cone, 2005*). *Mc4r* expressing neurons within the hypothalamus receive orexigenic and anorexigenic inputs from arcuate nucleus POMC and AgRP projections (*Balthasar et al., 2004*;

**eLife digest** Mutations in the gene that encodes a protein called the melanocortin-4 receptor are the most common genetic cause of early onset obesity in children. These mutations occur in about 1 in 1,500 people. The melanocortin-4 receptor is mostly found in the brain where it helps to balance how much a person eats with how many calories they burn. A mutation in just one of the two copies of the gene a person gets from their parents is enough to cause severe obesity.

Mice that have been genetically engineered to lack this gene develop all the same symptoms as humans with the mutation. These symptoms include early onset obesity, a slower than normal heart rate, and reduced activity in the nerves that communicate with many body tissues including the gut. Patients with this syndrome are less likely to develop obesity-linked high blood pressure, which could be considered protective from some of the ill effects of excess weight. As a result, studying the animal model of the syndrome may help scientists better understand why mutations in the gene for the melanocortin-4 receptor cause obesity and how to better care for people with these mutations.

Now, Litt et al. show that, contrary to expectations, mice lacking the gene for the melanocortin-4 receptor have a higher risk of heart failure than normal mice. An ultrasound scanner showed that the left side of the heart in the mice without the melanocortin-4 receptor becomes progressively larger and weaker. This reduces the heart's ability to pump blood. Additionally, Litt et al. showed that the energy-producing structures within cells, called mitochondria, are defective in the heart cells of these mice. These defects cause the mitochondria to work harder and produce more harmful byproducts. The mitochondria in the animal's muscles, however, appear normal.

Further experiments showed that the genes active in the hearts of the mice lacking melanocortin-4 receptors are similar to genes active in heart cells treated with doxorubicin, a cancer drug that is toxic to the heart. This drug is known to cause heart failure in some people. The experiments suggest that physicians should watch for signs of heart failure in people who have mutations that affect their melanocortin-4 receptors. Mice with one good copy of the gene did not have signs of heart failure, but they appeared more sensitive to the toxic affects of doxorubicin. These findings suggest that clinical studies are needed to determine if there are potential heart problems or drug sensitivities in patients with mutations that affect the melanocortin-4 receptors.

DOI: https://doi.org/10.7554/eLife.28118.002

*Balthasar et al., 2005*) and act to maintain energy homeostasis through modulation of both food intake (*Fan et al., 1997*; *Huszar et al., 1997*) and energy expenditure (*Ste Marie et al., 2000*). A reduction of MC4R signaling, through either genetic or pharmacological means, results in hyperphagia, bradycardia (*Wang et al., 2017*), and reduced blood pressure. Clinical studies have found that heterozygous *Mc4r* mutations confer protection from obesity-associated hypertension through reduced sympathetic tone (*Sweeney, 2010*; *Greenfield et al., 2009*; *Sayk et al., 2010*). Mice and humans with MC4R mutations also experience hyperinsulinemia that exceeds their degree of adiposity due to the role MC4R in the suppression of insulin release (*Fan et al., 2000*; *Mansour et al., 2010*). Furthermore, patients with *Mc4r* heterozygosity have reduced growth hormone suppression in response to obesity when compared to patients matched with standardized BMI (*Martinelli et al., 2011*). Therefore, while the effects of *Mc4r* deletion on peripheral vascular resistance may be cardio-protective, other aspects of the *Mc4r* obesity syndrome, such as hyperinsulinemia, incomplete growth hormone suppression, and altered autonomic tone are potentially cardiotoxic. Since no group to our knowledge has directly examined the effects of MC4R deletion on myocardial function, we chose to examine how *Mc4r* deletion affects myocardial function in vivo, and further characterized its effects on myocardial energy metabolism ex vivo.

## Results

### Cardiac function in heterozygous, and homozygous MC4R knockout mice

In order to characterize the effects of MC4R on heart function, age- and sex-matched *Mc4r−/−*, *Mc4r+/-* and WT mice were serially assessed using echocardiography. An age dependent cardiomyopathy was observed in male *Mc4r−/−* animals (*Figure 1A–B*), which included cardiac dilatation and reduced contractility. At 26 weeks of age, a significant reduction in fractional shortening (FS), a measure of myocardial contractility (*Figure 1C*) as well as ejection fraction (*Figure 1D*) could be observed. A significant increase in left ventricular diameter (LVIDd) was also observed in *Mc4r−/−* mice at 26 weeks of age, likely as a compensatory response to reduced contractility (*Figure 1E*). As has been shown previously (*Stepp et al., 2013*), a reduction in heart rate was also observed in the *Mc4r−/−* mice at later time points (*Figure 1F*). No significant change in heart wall thickness (LVPWd) was observed indicating the absence of pathological hypertrophy or atrophy (*Figure 1G*). Female *Mc4r−/−* mice were also examined with echocardiography. *Mc4r−/−* female mice displayed a similar phenotype to their male counterparts at 26 weeks (*Figure 1H–I*). This included a similar reduction of FS (*Figure 1J*) and EF (*Figure 1K*), an increase in LVIDd (*Figure 1L*) a reduction in heart rate (*Figure 1M*) and no significant reduction in LVPWd at 26 weeks (*Figure 1N*). ECG rhythm strips of male *Mc4r−/−* mice also revealed a bradycardic arrhythmia with dropped p waves at the 30 week time point (*Figure 2*).

Given these observations in male and female mice, we next sought to confirm that the cardiac deficits seen in the *Mc4r−/−* mouse were specific to the loss of *Mc4r* and not due to an anomalous background mutation. In order to accomplish this, the *Mc4r* loxTB mouse, a distinct *Mc4r* knockout model (*Balthasar et al., 2005*) was also examined by echocardiography. LoxTB *Mc4r* knockout animals displayed similar signs of myocardial dysfunction (*Figure 3A–B*) including reduced fractional shortening and ejection fraction with a trend towards increased LVIDd (*Figure 3C–E*) but did not have a significantly reduced heart rate, or heart wall thinning (*Figure 3F–G*). Together, these results demonstrate that *Mc4r* deletion leads to a progressive cardiomyopathy.

Both pre-clinical and epidemiological data support a causal role of hyperglycemia and insulin resistance in the development of heart failure (*Bugger and Abel, 2014*). Similarly, investigators have observed variable degrees of cardiomyopathy in mouse models of diet induced obesity (DIO) (*Battiprolu et al., 2012*; *Brainard et al., 2013*; *Calligaris et al., 2013*; *Heydemann, 2016*; *Raher et al., 2008*; *Wang et al., 2017*; *Christoffersen et al., 2003*; *Semeniuk et al., 2002*). The *Mc4r* is not expressed in adult mouse myocardial tissue (*Figure 4A–C*). Therefore, it is plausible that certain effects of the cardiomyopathy in *Mc4r−/−* mice are due directly to insulin resistance and/or negative metabolic effects of obesity. In order to address this possibility, a cohort of age matched diet induced obesity mice was generated and evaluated by echocardiography. Age matched male C57BL/6J wild type mice were placed on a 60% HFD for 35 weeks until they reached the same body weight as the *Mc4r−/−* animals (*Figure 5A*). Despite their similar body weights, DIO animals and *Mc4r−/−* animals had distinct body compositions, as previously described. When compared to DIO mice, *Mc4r−/−* mice accumulate more lean mass (*Figure 5B–C*) while DIO animals preferentially accumulate fat mass (*Figure 5D–E*). Despite these differences in body composition, *Mc4r−/−* and DIO animals had similar glucose tolerance in a glucose tolerance test when dosed in proportion to their lean mass (*Figure 5F–G*). Furthermore, there was no difference in myocardial insulin sensitivity between DIO and *Mc4r−/−* mice (*Figure 5H*) Thus, this cohort was appropriately matched for body weight and glucose tolerance. *Mc4r−/−* animals exhibited reduced cardiac contractility, while DIO and WT controls did not (*Figure 5I–J*). *Mc4r* knockouts had significantly lower FS and EF (*Figure 5K–L*) and significantly higher LVIDd (*Figure 5M*) while DIO and WT controls did not. In this cohort, *Mc4r−/−* males also displayed a trend towards a reduction in HR and LVPWd when compared to the DIOs (*Figure 5N–O*) however this difference did not reach statistical significance. These data demonstrate that the cardiomyopathy seen in *Mc4r−/−* is caused by the loss of MC4R function, rather than a secondary effect of the obesity caused by loss of the MC4R.

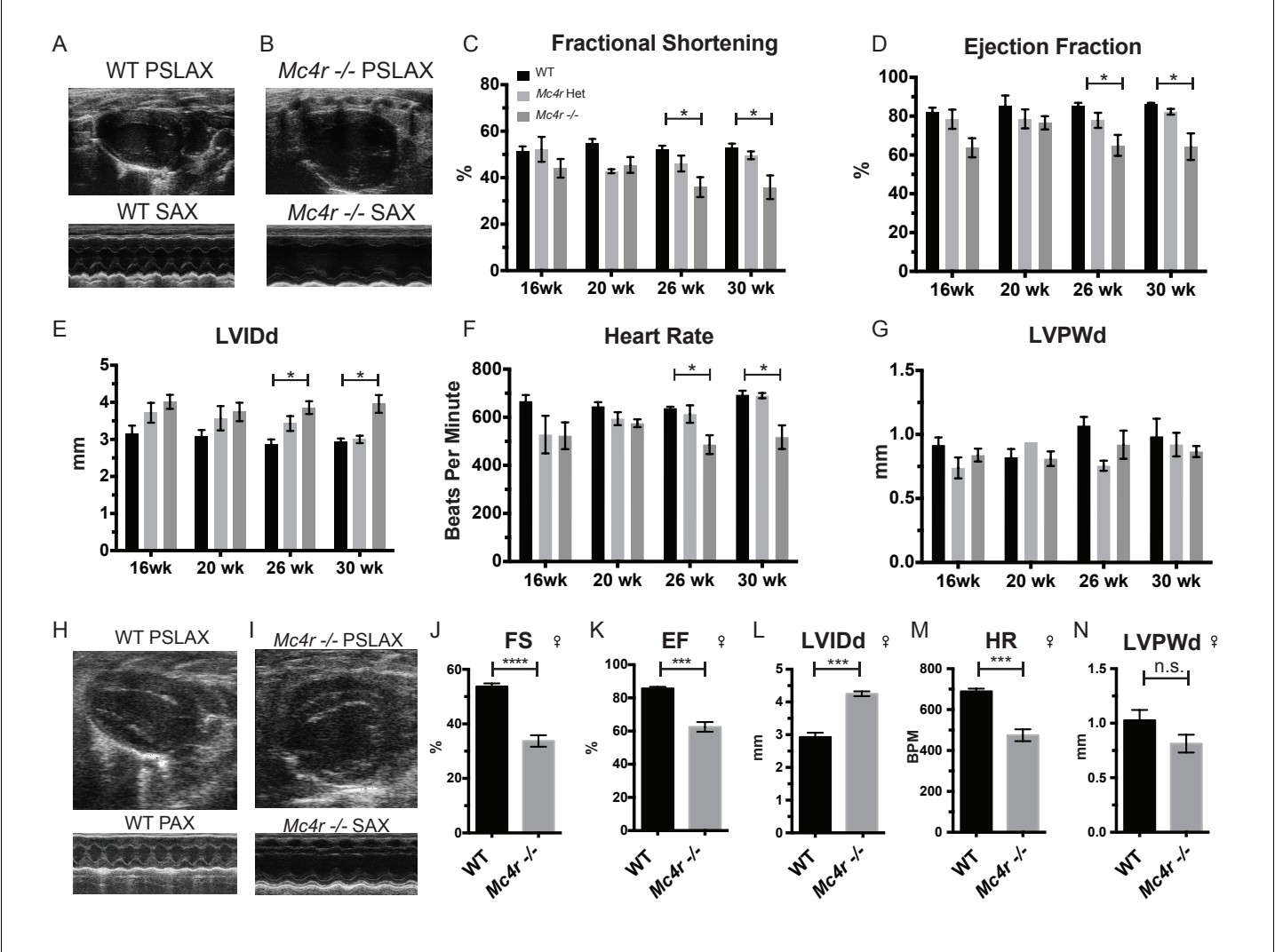

**Figure 1.** *Mc4r* deletion causes progressive cardiac dilatation and reduced contractility. (**A**) Representative B-mode Parasternal long axis (PSLAX) image at diastole (top) and M-mode short axis (SAX) image of male WT C57B6/J mouse at 26 weeks of age. (**B**) Representative B-mode PSLAX image at diastole (top) and M-mode SAX image of male *Mc4r−/−* mouse 26 weeks of age. (**C**) Fractional Shortening (FS) of male WT, *Mc4r−/+*, and *Mc4r−/−* mice at 16, 20, 26 and 30 weeks of age. (2-way ANOVA Sidak multiple comparison test, p<0.05, n = 5–6) (**D**) Ejection Fraction (2-way ANOVA Sidak multiple comparison test, p<0.05, n = 5–6) (**E**) Left ventricular internal dimension during diastole (LVIDd) (2-way ANOVA Sidak multiple comparison test, p<0.05, n = 5–6) (**F**) Heart Rate (2-way ANOVA Sidak multiple comparison test, p<0.05, n = 5–6) (**G**) Left ventricle posterior wall thickness during diastole (LVPWd) (2-way ANOVA Sidak multiple comparison test, p>0.05, n = 5–6) (**H**) Representative B-mode PSLAX image at diastole (top) and M-mode SAX image of female WT C57B6/J mouse at 26 weeks of age. (**I**) Representative B-mode PSLAX image at diastole (top) and M-mode SAX image of female *Mc4r−/−* C57B6/J mouse 26 weeks of age. (**J**) Fractional shortening of female WT and *Mc4r−/−* mice at 26 weeks. (Student's t-test p<0.0001, n = 5) (**K**) Ejection fraction of female WT and *Mc4r−/−* mice at 26 weeks. (Student's t-test p<0.001, n = 5) (**L**) LVIDd (Student's t-test p<0.0001, n = 5) (**M**) HR (Student's t-test p<0.001, n = 5–6) (**N**) LVPWd (Student's t-test p=0.1185, n = 5 ).

DOI: https://doi.org/10.7554/eLife.28118.003

The following source data is available for figure 1:

**Source data 1.** Source data for *Figure 1C–G and J-N*.
DOI: https://doi.org/10.7554/eLife.28118.004

## Effect of *Mc4r* deletion on mitochondrial structure and function

*Mc4r−/−* myocardium was examined to determine if the tissue displayed signs of heart failure, such as interstitial fibrosis or lipid deposition. *Mc4r−/−* hearts appeared grossly larger (*Figure 6A*) and displayed a significantly higher cardiac wet weight when compared to WT controls at 30 weeks of age (*Figure 6B*). At this time point there was no difference in the heart weight to lean body mass

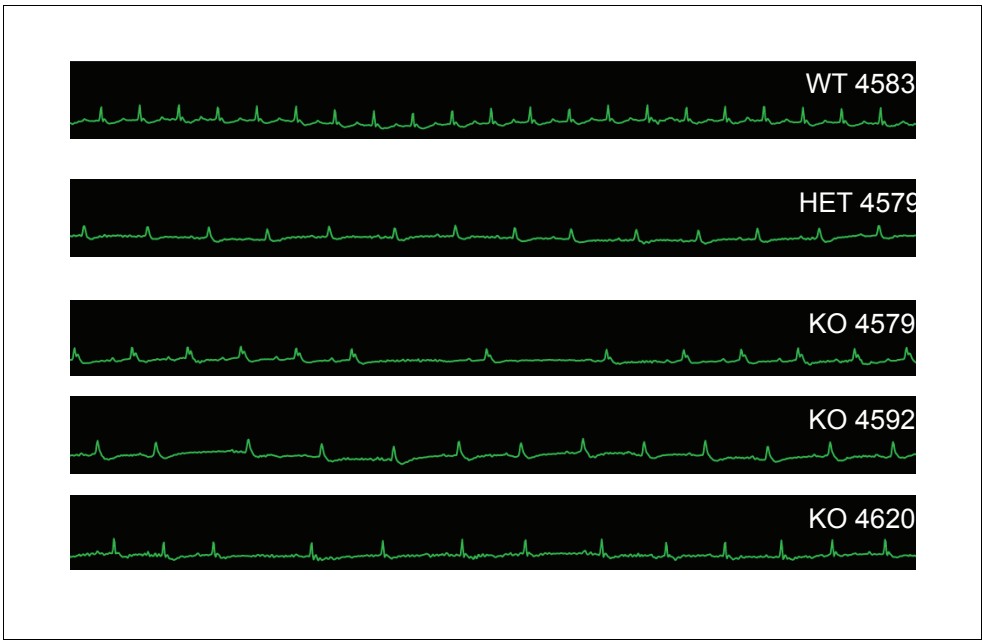

**Figure 2.** *Mc4r−/−* animals display bradycardic arrhythmias. Rhythm strips of 30 week old WT, *Mc4r-/+* and *Mc4r−/−* animals taken during echocardiograms. *Mc4r−/−* mice display sinus bradycardia with abnormal rhythmicity.

DOI: https://doi.org/10.7554/eLife.28118.005

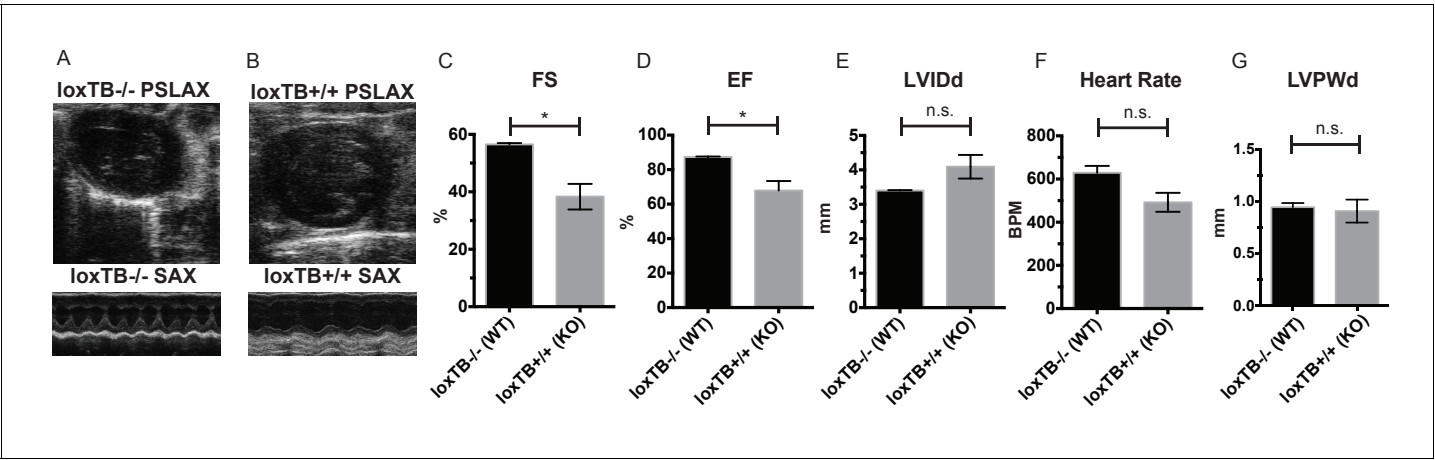

**Figure 3.** *Mc4r* loxTB animals phenocopy the reduced cardiac function seen in *Mc4r−/−* mice. (**A**) Representative B-mode PSLAX image at diastole (top) and M-mode SAX image of male *Mc4r* loxTB -/- (WT) mouse at 35 weeks of age. (**B**) Representative B- mode PSLAX image at diastole (top) and M-mode SAX image of male *Mc4r* loxTB +/+ (*Mc4r−/−*) mouse 35 weeks of age. (**C**) Fractional Shortening (FS) of male *Mc4r* loxTB -/- and *Mc4r* loxTB +/+mice at 35 weeks (Student's t-test, p<0.05, n = 5) (**D**) Ejection Fraction (Student's t-test, p<0.05, n = 5). (**E**) Left ventricular internal dimension; diastole (LVIDd) (Student's t-test, p=0.0838, n = 5). (**F**) Heart Rate (Student's t-test, p>0.05, n = 5). (**G**) Left ventricle posterior wall thickness; diastole (Student's t-test, p<0.05, n = 5).

DOI: https://doi.org/10.7554/eLife.28118.006

The following source data is available for figure 3:

**Source data 1.** Source data for *Figure 3C–G*.

DOI: https://doi.org/10.7554/eLife.28118.007

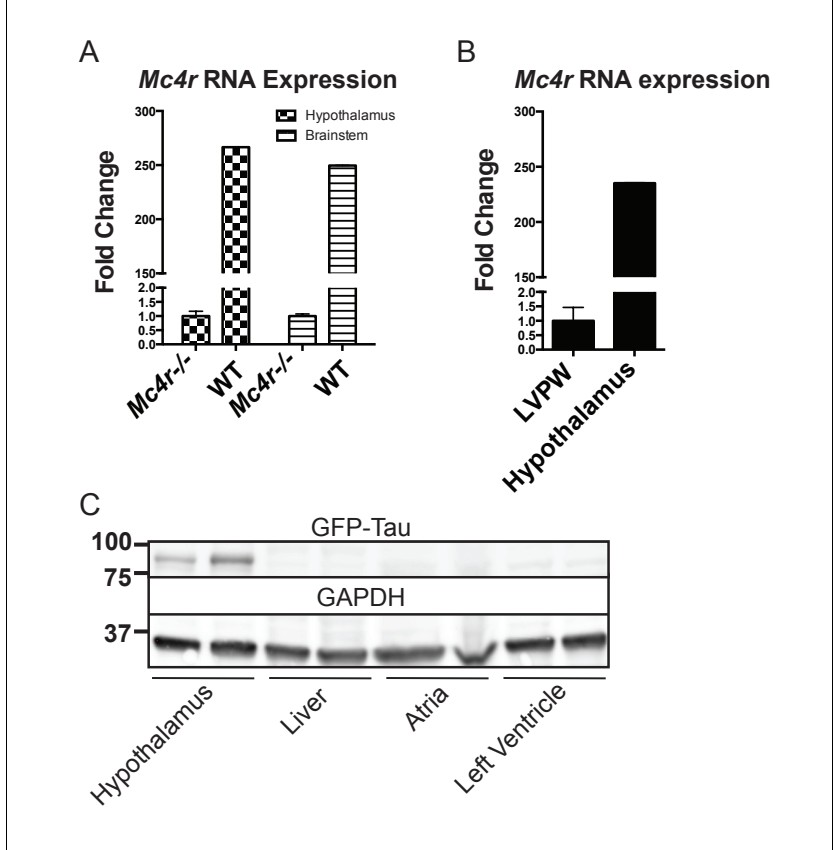

**Figure 4.** *Mc4r* is not expressed in adult heart tissue. (**A**) qRT-PCR comparison of *Mc4r* expression in heart and hypothalamus using the ΔΔCT method. (Student's t-test, p<0.0001, n = 8) (**B**) qRT-PCR comparison of *Mc4r* expression in the hypothalamus and brainstem between *Mc4r−/−* and WT tissue demonstrating dynamic range of qRT-PCR primers (Student's t-test, p<0.0001, n = 8). (**C**) Representative western blot of *Mc4r* -Tau-Sapp hypothalamic, hepatic, atrial and ventricular lysates.

DOI: https://doi.org/10.7554/eLife.28118.008

The following source data is available for figure 4:

**Source data 1.** Source data for *Figure 4A–B*.
DOI: https://doi.org/10.7554/eLife.28118.009

(HW/LBM) ratio (*Figure 6C*), likely due to the generalized increase in lean mass seen in *Mc4r−/−* mice. However, the heart weight of *Mc4r−/−* mice was significantly higher than that of age matched DIO animals (*Figure 6D*). Histological analysis of *Mc4r−/−* tissue by H&E stain was unremarkable and did not reveal signs of an overt pathological insult such as anoxia, inflammatory infiltration or fibrosis (*Figure 6E*). Examination of H&E sections at 500 µm intervals further revealed that *Mc4r−/−* mice did not displayed myocyte hypertrophy (*Figure 6F*). Similarly, lipid deposition was not observed by Oil Red O staining (*Figure 6G*). However, transmission electron microscopy (TEM) showed ultrastructural signs of heart failure. TEM imaging of myocardium in *Mc4r−/−* mice revealed mitochondrial pleomorphy and cardiomyocyte dropout (*Figure 6H–I*). Based on these findings, we hypothesized that *Mc4r−/−* animals were experiencing mitochondrial dysfunction.

In order to determine if *Mc4r−/−* mice had alterations in mitochondrial number, tissue samples were examined for mitochondrial DNA content using qPCR. With this assay, there was no difference in mitochondrial DNA content relative to genomic DNA content in male *Mc4r−/−* myocardium (*Figure 7A*). Accordingly, there was no difference in the abundance of electron transport chain (ETC) complex I-IV proteins or ATP synthase by western blot (*Figure 7B–C*). Based on these results and the abnormal mitochondrial morphology on TEM, we then sought to characterize *Mc4r−/−* myocardial mitochondrial function in situ using high-resolution respirometry of saponin permeabilized left

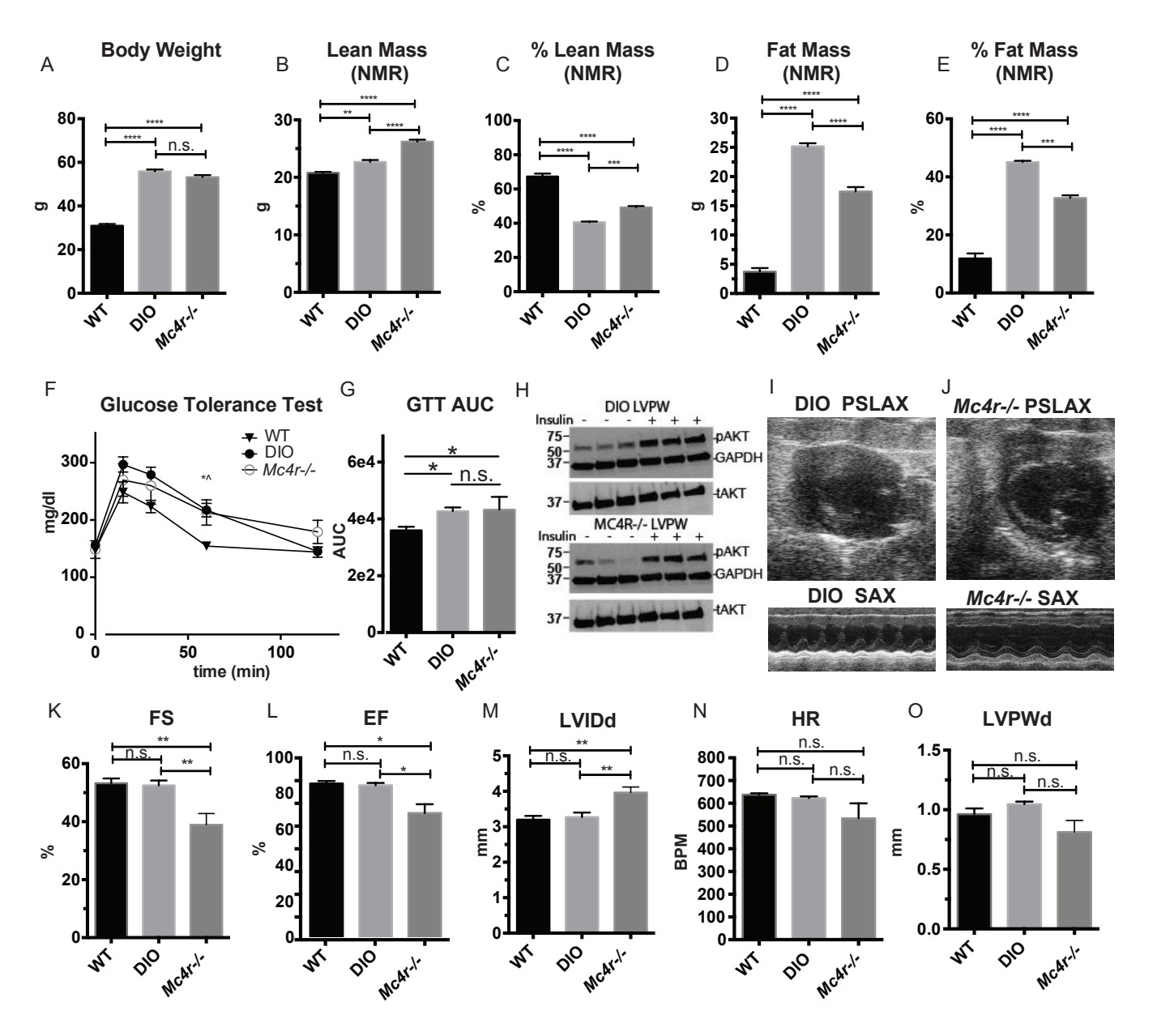

**Figure 5.** Age and weight matched wild type animals fed a high fat diet do not phenocopy the cardiomyopathy seen in *Mc4r−/−* mice. (A) Body weights of age matched WT, diet induced obese (DIO), and *Mc4r−/−* mice. (1-way ANOVA Sidak multiple comparison test, p<0.0001, n = 5). (B) Lean mass of WT, DIO and *Mc4r−/−* mice by NMR (1-way ANOVA Sidak multiple comparison, p<0.01, n = 5) (C) Percent Lean Mass comparison of WT, DIO and *Mc4r−/−* mice (1-way ANOVA Sidak multiple comparison, p<0.001, n = 5) (D) Fat mass by NMR (1-way ANOVA Sidak multiple comparison, p<0.0001, n = 5) (E) Percent Fat Mass (1-way ANOVA Sidak multiple comparison, p<0.0001, n = 5) (F) Glucose tolerance test time course. 2 mg glucose per kg lean mass (2-way ANOVA Sidak multiple comparison test, p<0.05, n = 5) (G) Area under the curve of glucose tolerance test (Dunn's multiple comparison test, p<0.05, n = 5) (H) pAkt T-308 blots of the LVPW in *Mc4r−/−* and DIO mice following the injection of saline or 1 U/kg insulin. (I) Representative B-mode PSLAX image at diastole (top) and M-mode SAX image of male DIO mice at 30 weeks of age. (J) Representative B-mode PSLAX image at diastole (top) and M-mode SAX image of male *Mc4r−/−* mouse 30 weeks of age. (K) Fractional Shortening of WT, DIO and *Mc4r−/−* mice (1-way ANOVA Sidak multiple comparison, p<0.01, n = 5) (L) Ejection Fraction of WT, DIO and *Mc4r−/−* mice (1-way ANOVA Sidak multiple comparison, p<0.05, n = 5) (M) LVIDd (1-way ANOVA Sidak multiple comparison, p<0.01, n = 5) (N) HR (1-way ANOVA Sidak multiple comparison, p>0.05, n = 5) (O) LVPWd (1-way ANOVA Sidak multiple comparison, p>0.05, n = 5).

DOI: https://doi.org/10.7554/eLife.28118.010

The following source data is available for figure 5:

*Figure 5 continued on next page*

*Figure 5 continued*

**Source data 1.** Source data for *Figure 5A–G and K-O*.

DOI: https://doi.org/10.7554/eLife.28118.011

ventricle fibers. Using the protocol described in *Figure 7D*, no difference in the $O_2$ consumption between genotypes (*Figure 7E*) was observed upon exposure to glutamate and malate in the

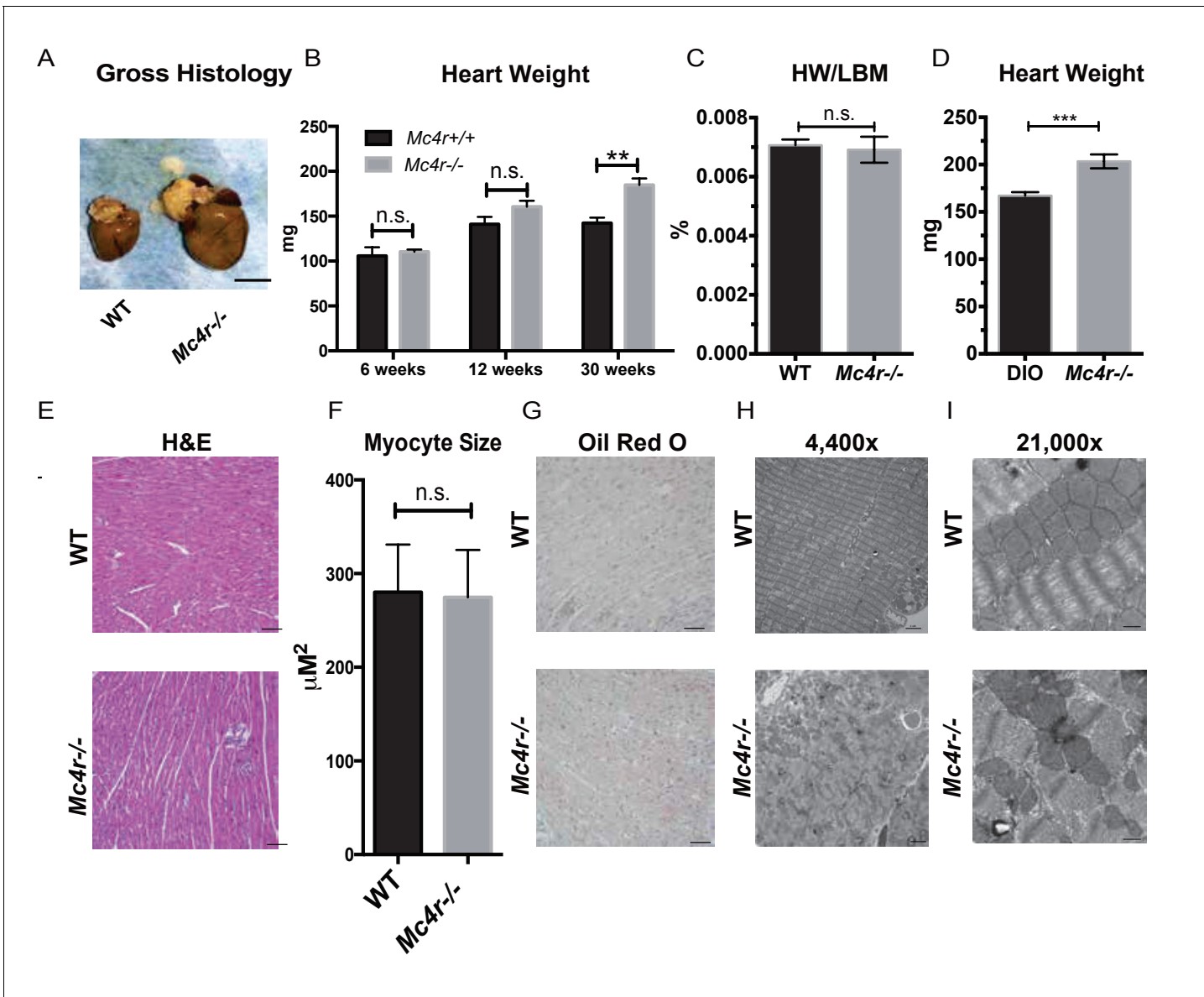

**Figure 6.** Histological analysis of *Mc4r−/−* hearts reveals mitochondrial pleomorphy and cardiomyocyte dropout. (A) Representative gross pathology of WT and *Mc4r−/−* hearts (B) Heart wet weight of 6, 12 and 30 week old WT and *Mc4r−/−* animals (Student's t-test, p<0.01, n = 4–7) (C) Heart weight to total lean mass ratio (Student's t-test, p>0.05, n = 5) (D) Heart wet weight of 40 week old *Mc4r−/−* and DIO animals. (Student's t-test; p<0.001, n = 7–8). (E) Representative H and E images of WT and *Mc4r−/−* myocardium; scale bar = 500 μm (F) Cross sectional myocyte area by H and E stain. (Student's t-test, p>0.05, n = 3 mice x three cross sections). (G) Representative Oil Red O images of WT and *Mc4r−/−* myocardium; scale bar = 500 μm (H-I) Representative TEM images WT and *Mc4r−/−* myocardium; 4400x scale bar = 2 μm; 21,000x scale bar = 500 nm.

DOI: https://doi.org/10.7554/eLife.28118.012

The following source data is available for figure 6:

**Source data 1.** Source data for *Figure 6B–D and 6F*.

DOI: https://doi.org/10.7554/eLife.28118.013

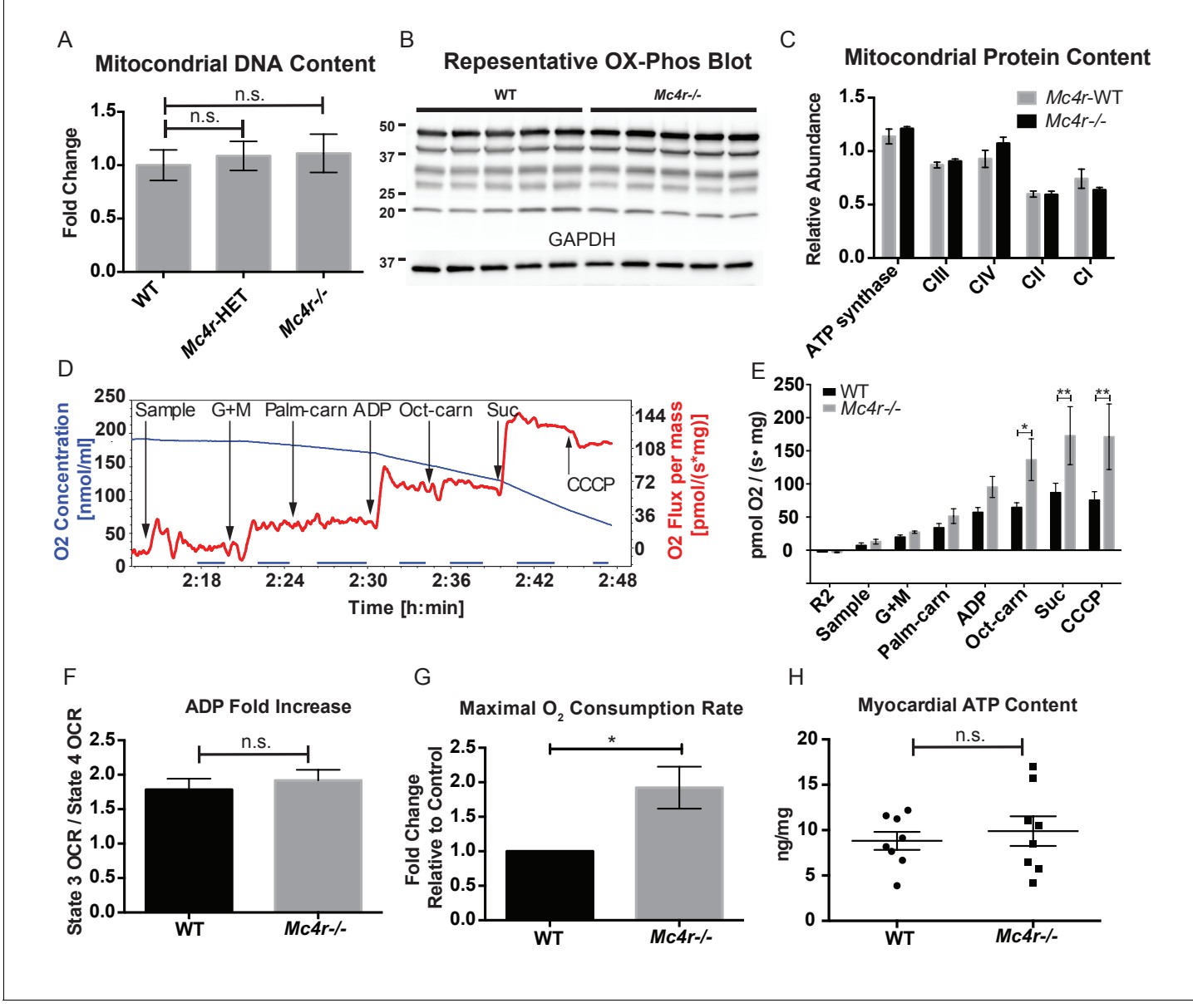

**Figure 7.** *Mc4r−/−* heart tissue displays increased mitochondrial capacity. (A) Relative qPCR of mitochondrial DNA content using H19 and CytB primers (Students t-test of ΔCT values, p>0.05, n = 6). (B) Representative western blot of whole cell tissue lysates from WT and *Mc4r−/−* myocardium of electron transport chain (ETC) proteins. (C) Relative quantification of ETC blot normalized to GAPDH content (2 way ANOVA Sidak multiple comparison test, p>0.05, n = 3 blots x 5 samples per genotype) (D) Representative trace of a respirometry assay. Chamber $O_2$ content is represented in blue while the rate of $O_2$ consumption per mg tissue is in red. The sampling time used for quantification is displayed in the blue bar on the x-axis and is obtained following stabilization of $O_2$ consumption rate in the chamber. (E) $O_2$ consumption rate comparison between permeabilized WT and *Mc4r−/−* myocardial tissue from 30 to 33 week old mice. (2 way ANOVA Sidak post test, p<0.05, n = 5) (F) ADP dependent $O_2$ consumption ratio (paired t-test, p>0.05, n = 5) (G) Ratio $O_2$ consumption of *Mc4r−/−* tissue compared to WT (ratio paired t-test, p<0.05, n = 5) (H) ATP content in WT and *Mc4r−/−* myocardial tissue (Student's t-test, p>0.05, n = 9).

DOI: https://doi.org/10.7554/eLife.28118.014

The following source data is available for figure 7:

**Source data 1.** Source data for *Figure 7A and C*, and 7E-H.

DOI: https://doi.org/10.7554/eLife.28118.015

absence of ADP (complex I substrates, state IV respiration). Similarly, addition of palmitoylcarnitine in the absence of ADP (Fatty Acid Substrate, state IV respiration) did not result in any difference in

O₂ consumption. When ADP was added to this reaction (Complex I substrates, state III respiration) there was a trend towards *Mc4r−/−* myocardium consuming more O₂ than control tissue. The increase in state III respiration became more pronounced in the presence of medium chain fatty acid substrates (L-Octanoylcarnitine), as well as in the presence of succinate (complex II substrate, state III mediated respiration). This increase in ETC capacity persisted when the ionophore CCCP was titrated into the reaction. Importantly, the fold increase upon ADP exposure was not different between samples indicating a reliable tissue preparation (*Figure 7F*). This demonstrates that 30-week-old *Mc4r−/−* myocardium displayed a 2-fold increase in total respiratory capacity (*Figure 7G*) without a subsequent increase in tissue ATP content (*Figure 7H*). These findings are in contrast to what is generally seen in dilated cardiomyopathy and right heart failure (*Talati et al., 2016*) but more closely mimics what has been observed in hypertrophic cardiomyopathy (*Rosca et al., 2013*).

In order to see if the mitochondrial phenotype precedes heart failure as well as the *Mc4r−/−* obesity phenotype, the O₂ consuming capacity of young lean *Mc4r−/−* myocardium was then analyzed using high-resolution respirometry. Similar to the older cohort, young lean *Mc4r−/−* animals display increased ADP dependent respiration and ETC capacity without any changes in ADP independent respiration (*Figure 8A*). Similar to the older animals, there was no difference in ADP fold change

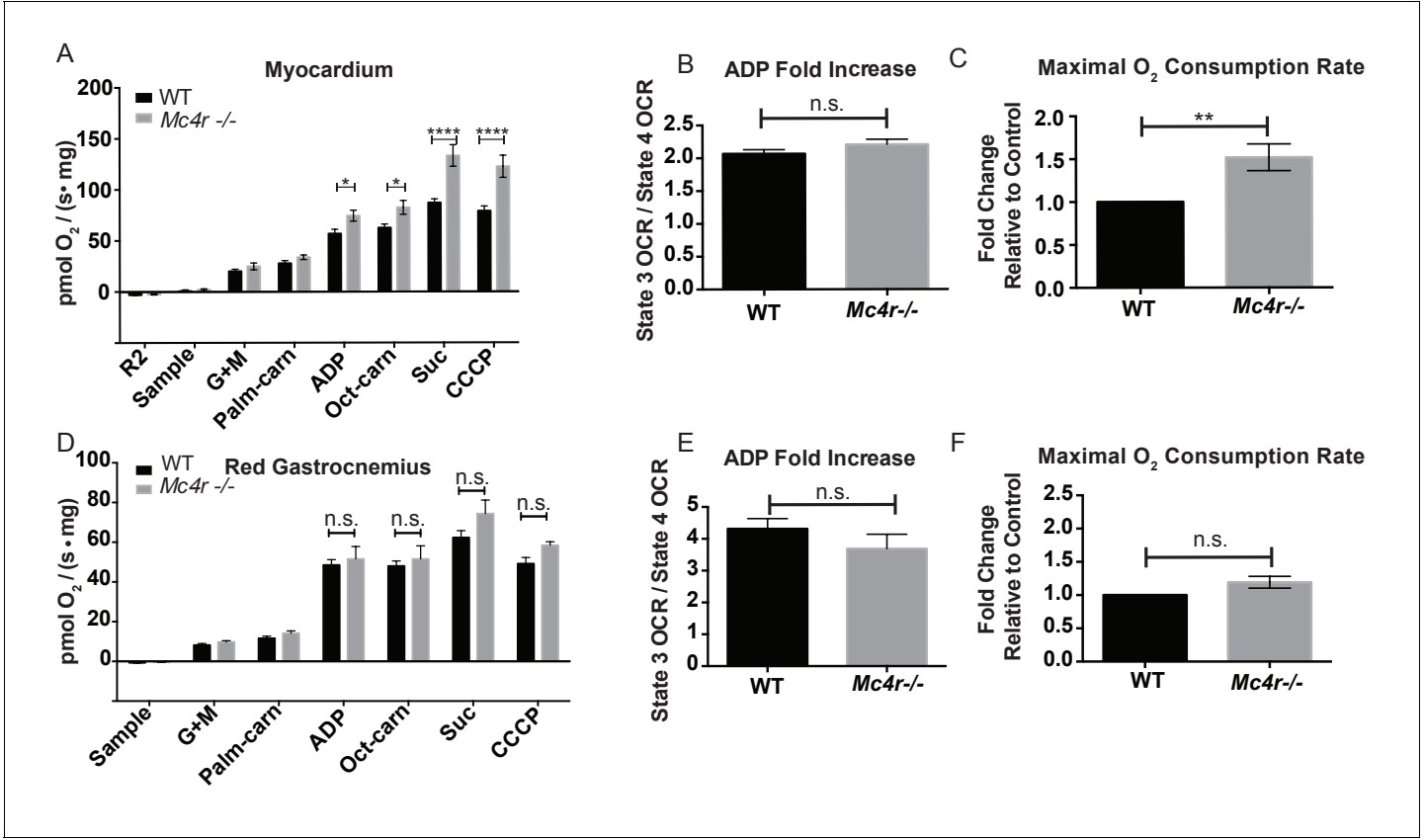

**Figure 8.** Mitochondrial capacity changes in *Mc4r−/−* animals are present in young animals and specific to the heart. (A) O₂ consumption rate comparison between WT and *Mc4r−/−* myocardial tissue from 6 to 12 week old mice. (2 way ANOVA Sidak post test, $p<0.05$, n = 7) at 6–12 weeks of age (B) ADP dependent O₂ consumption ratio (paired t-test, $p>0.05$, n = 7) (C) Ratio O₂ consumption of *Mc4r−/−* tissue compared to WT (ratio paired t-test, $p<0.05$, n = 7) (D) O₂ consumption rate comparison between WT and *Mc4r−/−* red gastrocnemius tissue from 6 to 12 week old mice. (2 way ANOVA Sidak post test, $p>0.05$, n = 6) at 6–12 weeks of age (E) ADP dependent O₂ consumption ratio of red gastrocnemius tissue from 6 to 12 week old mice (paired t-test, $p>0.05$, n = 6) (F) Ratio O₂ consumption of *Mc4r−/−* red gastrocnemius tissue compared to WT red gastrocnemius tissue (ratio paired t-test, $p>0.05$, n = 6).

DOI: https://doi.org/10.7554/eLife.28118.016

The following source data is available for figure 8:

**Source data 1.** Source data for *Figure 8A–F*.
DOI: https://doi.org/10.7554/eLife.28118.017

between *Mc4r−/−* and WT cardiac tissue (*Figure 8B*). Overall, lean *Mc4r−/−* myocardial tissue displayed a 1.5 increase in $O_2$ consuming capacity compared to WT controls (*Figure 8C*). In order to determine if this change is specific to heart tissue, permeabilized red gastrocnemius muscle fibers were then examined with respirometry. Unlike the myocardial tissue, skeletal muscle $O_2$ consumption capacity in lean *Mc4r−/−* animals was not different from controls with respect to both ADP dependent and independent respiration (*Figure 8D–F*). Furthermore, no significant difference in ETC complex proteins was detected in muscle fibers (*Figure 9A–B*). Thus, the observed increase in myocardial oxygen consumption in *Mc4r−/−* mice is specific to the myocardium, independent of obesity, and precedes the development of heart failure.

## Potential mechanisms underlying cardiomyocyte dysfunction in *Mc4r* knockout mice

Based on these observations, we then sought to understand how an increase in $O_2$ consumption without an increase in ATP content might be contributing to the observed defect in myocardial contractility. One possible explanation is the excessive production of reactive oxygen species (ROS). Formed as a byproduct of normal aerobic metabolism, ROS play both physiological and pathophysiological roles throughout the body. However, excessive ROS production is known to contribute to heart failure through irreversible modifications of cellular lipids, proteins and DNA (*Giordano, 2005*). In order to study ROS levels in *Mc4r−/−* myocardium, tissue lysates were examined using the 2',7' − dichlorofluorescein diacetate (DCFDA) oxidation assay. When compared to WT tissue, an increase in oxidized 2', 7' −dichlorofluorescein (DCF) was observed in both young *Mc4r−/−* (*Figure 10A*) and old *Mc4r−/−* (*Figure 10B*) myocardium. Together with the respirometry data (*Figure 7D–F*, *Figure 8A–C*) and ATP measurements (*Figure 7G*) this finding suggests that increased $O_2$ consumption in myocytes of *Mc4r−/−* mice does not lead to increased ATP production but rather the production of ROS. Since ROS can be both physiological and pathophysiological, pathological intermediates and end products of ROS were then examined. While no significant increase in malondialdehyde, an intermediate of lipid peroxidation, was found (*Figure 10C*), a significant increase in 4-hydroxynonenal protein adducts was observed (*Figure 10D*). 4-HNE adducts result from irreversible lipid-protein covalent bonds caused by excessive ROS, and have been shown to be a causative mechanism for ROS mediated tissue damage (*Mali and Palaniyandi, 2014*).

After identifying ROS as a link between increased $O_2$ consumption and tissue damage, we next sought to understand how loss of MC4R leads to this pathological insult. MC4R is primarily expressed in the central and peripheral nervous system. Previous studies have been unable to detect *Mc4r* expression in mouse myocardium. As described above, qRT-PCR was used to characterize the expression pattern of *Mc4r*. Similar to previous studies, a nearly 250 fold enrichment of *Mc4r* mRNA was observed in WT hypothalamus and brain stem versus hypothalamus or brainstem from *Mc4r−/−* tissue (*Figure 4*). However, no significant fold change in *Mc4r* mRNA was observed in the left ventricle between WT

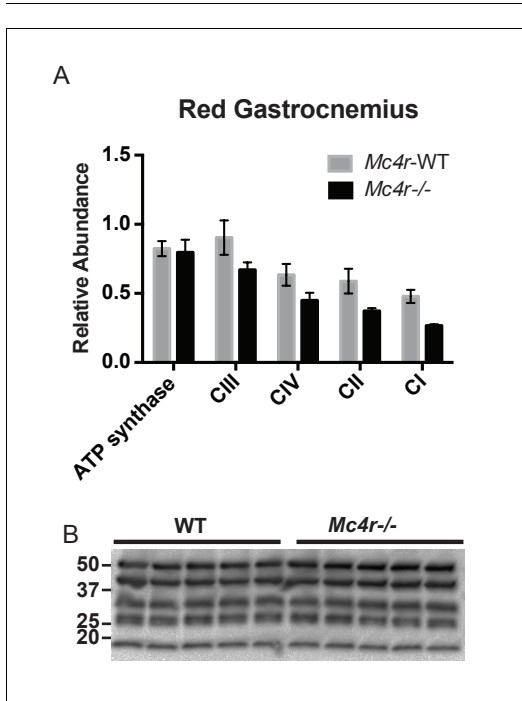

**Figure 9.** Red Gastrocnemius Ox-Phos components. (**A**) Relative quantification of ETC blot normalized to ATP synthase content in WT tissue (2 way ANOVA Bonferroni post test, p>0.05, n = 3 blots x 5 samples per genotype) (**B**) Representative western blot from tissue lysates from WT and *Mc4r−/−* myocardium of ETC proteins.
DOI: https://doi.org/10.7554/eLife.28118.018
The following source data is available for figure 9:

**Source data 1.** Source data for *Figure 9A*.
DOI: https://doi.org/10.7554/eLife.28118.019

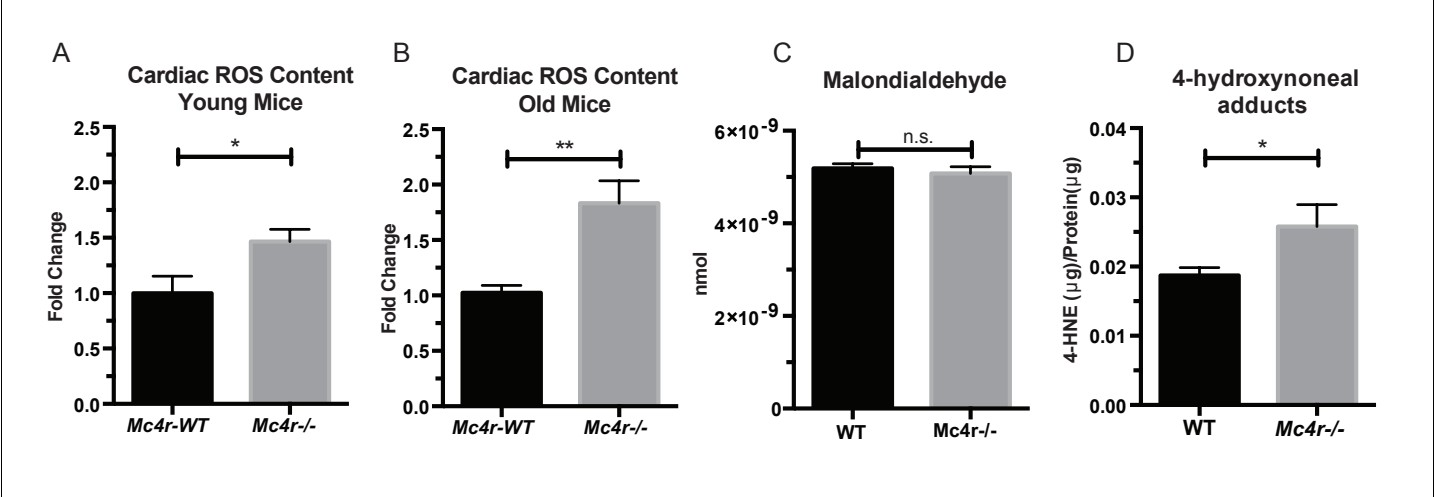

**Figure 10.** *Mc4r−/−* hearts display signs of increased ROS production. (**A**) DCF assay comparing myocardial lysates from 6 to 12 week old *Mc4r−/−* and WT mice (ratio paired t-test, p<0.05, n = 5). (**B**) DCF assay comparing myocardial lysates from 30 to 33 week old *Mc4r−/−* and WT mice (ratio paired t-test, p<0.05, n = 5). (**C**) Malondialdehyde content in 30 week old *Mc4r−/−* and WT myocardium (Student's t-test, p>0.05, n = 5) (**D**) 4-HNE protein content per mg total protein (Student's t-test, p>0.05,n = 5).

DOI: https://doi.org/10.7554/eLife.28118.020

The following source data is available for figure 10:

**Source data 1.** Source data for *Figure 10A–D*.
DOI: https://doi.org/10.7554/eLife.28118.021

and *Mc4r−/−* mice (*Figure 4B*). Furthermore, we did not detect myocardial GFP expression in *Mc4r-Sapp* animals (*Liu et al., 2003*) in either the atrial or ventricular tissue (*Figure 4C*). Based on these results, loss of *Mc4r* appears to cause cardiomyopathy through an indirect mechanism.

In order to determine a signaling pathway that facilitates this indirect mechanism, *Mc4r−/−* myocardium of 30 week old mice was compared to age matched control myocardium using RNA-seq. Similar to the qRT-PCR and western blot data, *Mc4r* expression in the myocardium was undetectable. However, 247 transcripts that were significantly different from WT controls (*Figure 11A*) were identified. Differential expression of select genes was confirmed using qRT-PCR including increased expression of *Myl7* and reduced expression of *Ppargc1a* (*Figure 11B*). In order to understand the significance of these gene changes, gene set enrichment analysis (GSEA) was performed to identify a known stimulus that promotes a similar transcriptional change. GSEA revealed that *Mc4r−/−* gene changes in the myocardium resemble that seen following doxorubicin treatment — a known inducer of ROS and heart failure (*Figure 11C*). Pathway ontology analysis of all significantly different transcripts was then used to identify a potential mechanism responsible for heart failure. This analysis revealed a contribution 45 known signaling pathways and protein classes (*Figure 11D*) including growth factor signaling, GPCR signaling, cytoskeleton regulation, glycolysis and inflammation. Thus, while the pathway changes associated with *Mc4r* deficiency appear to affect multiple nodes of cardiac tissue function, these changes appear to induce a tissue phenotype that is consistent with oxidative stress.

The data described above suggest that MC4R may somehow play a protective role in preventing cardiac ROS stress. In order to develop a model system for testing this, we decided to study anthracycline-induced cardiotoxicity in a condition in which MC4R signaling could still be modulated. *Mc4r +/−*animals and their WT siblings were injected with a sub-cardiotoxic dose of DOX (2 × 5 mg/kg IP injections) (*Figure 12A*). Body weight measurements reveal that DOX injected *Mc4r+/−*mice lost nearly twice as much weight as their DOX injected WT controls (*Figure 12B*), suggesting that even partial reduction of MC4R signaling may increase sensitivity to cardiotoxic insults.

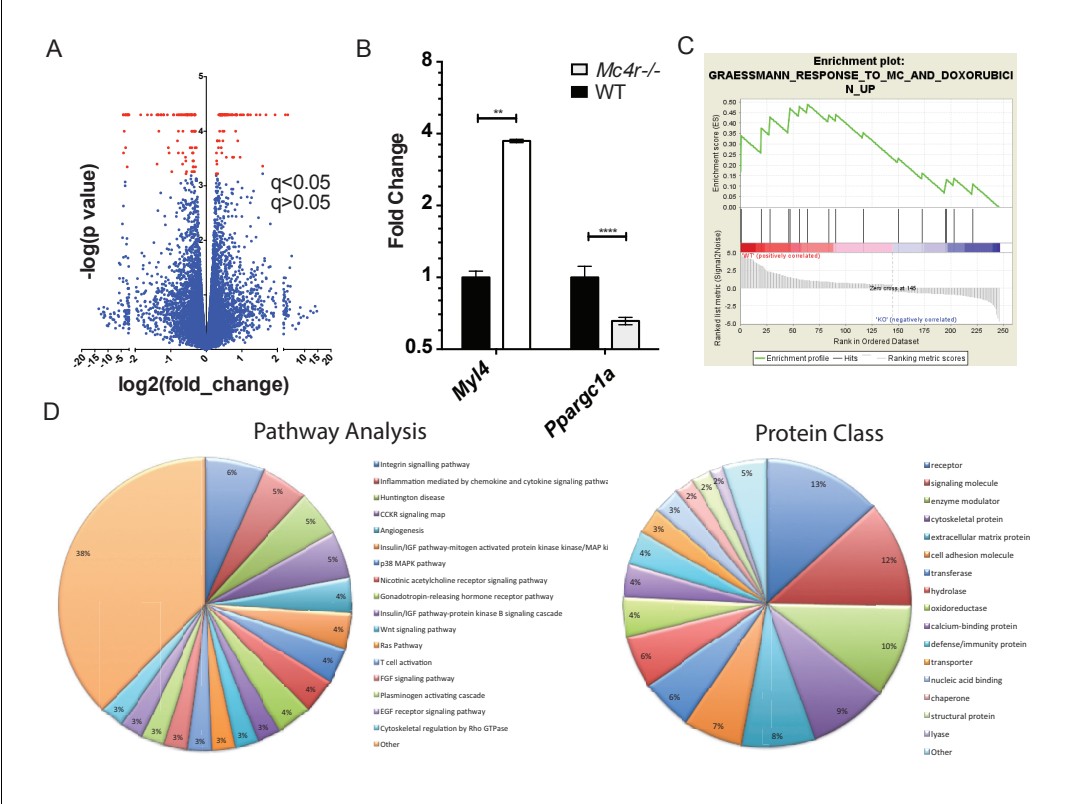

**Figure 11.** Gene changes in *Mc4r−/−* myocardium. (**A**) Volcano plot of RNA-seq results. Red points indicate gene changes that reached statistical significance (n = 4 WT, n = 6 *Mc4r−/−*). (**B**) qRT-PCR confirmation of select gene changes found with RNA-seq (Student's t-test, p<0.01, n = 4 WT, n = 6 *Mc4r−/−*) (**C**) Gene set enrichment analysis (GSEA) results from a different gene transcripts found using RNA-seq (nominal p<0.001). (**D**) Gene ontology pathway analysis of significantly unregulated genes using PANTHER Pathways Overrepresentation test.

DOI: https://doi.org/10.7554/eLife.28118.022

The following source data is available for figure 11:

**Source data 1.** Source data for *Figure 11B*.

DOI: https://doi.org/10.7554/eLife.28118.023

## Discussion

In this report, we describe the development of cardiomyopathy in *Mc4r−/−* mice. In this mouse model of obesity, we observed a progressive decline in contractility as well as an increase in cardiac chamber size. This decline in cardiac function is found in *Mc4r−/−* animals regardless of animal sex or knockout strategy but is absent in weight matched, diet induced obese mice. Using transmission electron micrographs, we observed grossly disorganized myofibers and pleomorphic mitochondria in *Mc4r−/−* myocardium. Subsequent functional testing of myocardium revealed a nearly 2-fold increase in $O_2$ consuming capacity in 30-week-old *Mc4r−/−* myocardium as well as a 2-fold increase in ROS. Studies in young lean *Mc4r−/−* myocardium revealed similar findings with a 1.5-fold increase in $O_2$ consuming capacity and a 1.5-fold increase in ROS. We did not detect any change in skeletal muscle $O_2$ consuming capacity indicating that this change is specific to the myocardium. The underlying pathology appears to be due to an indirect mechanism as we were unable to detect *Mc4r* expression in atrial or ventricular tissue (*Figure 4A–C*). Subsequent in silico analysis of RNAseq revealed gene set similarities in *MC4R−/−* myocardium to those seen in doxorubicin treated myocardial cells (*Figure 11*). Based on these findings, *Mc4r+/-*and WT mice were treated with a low dose doxorubicin treatment to see if a partial reduction of MC4R signaling might cause an increased sensitivity to a cardiotoxic challenge. These studies revealed that *Mc4r+/-*animals were more sensitive to doxorubicin than their WT controls (*Figure 12*), as indicated by an increased cachexigenic response. This was quite surprising, because inhibition of MC4R signaling is well documented to

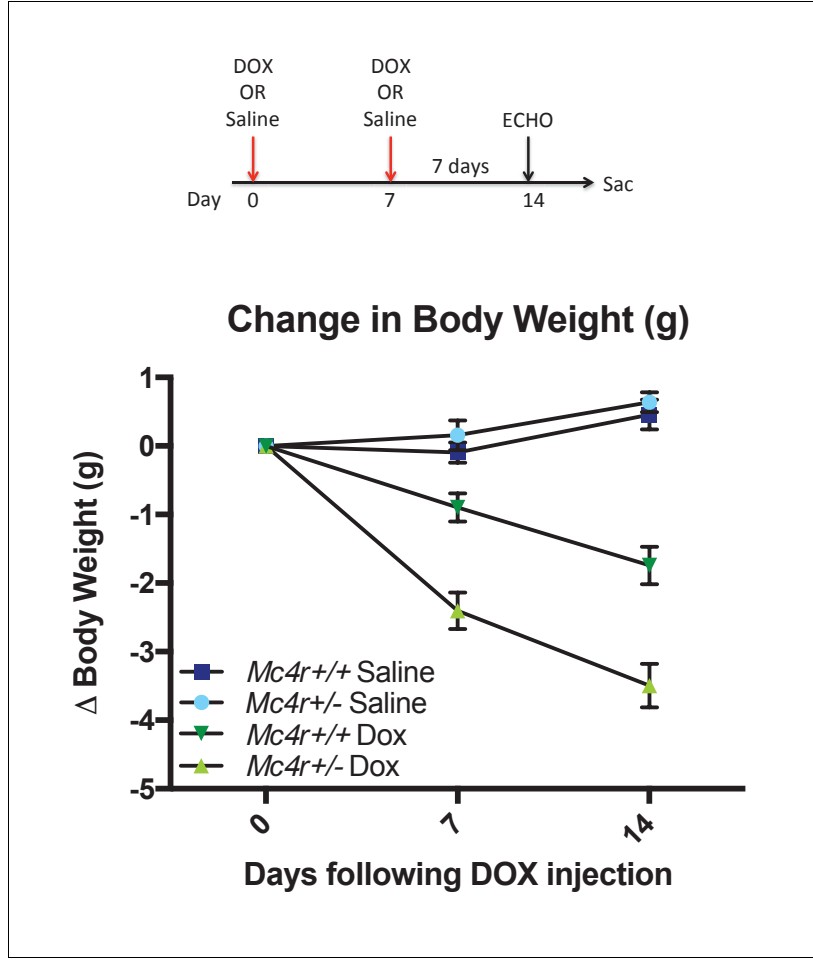

**Figure 12.** *Mc4r* ± mice are hypersensitive to doxorubicin. (**A**) Treatment strategy to subclinical cardiac dysfunction in WT C57B6/J mice. Mice were given 5 mg/kg doxorubicin over the course of 2 weeks during which time their bodyweights were recorded. (**B**) Body weights of *Mc4r+/-* and WT during the course of the treatment paradigm. *Mc4r+/-* mice almost twice as much weight as their WT controls. (One Way ANOVA, p<0.05, n = 6–14).
DOI: https://doi.org/10.7554/eLife.28118.024
The following source data is available for figure 12:

**Source data 1.** Source data for *Figure 12B*.
DOI: https://doi.org/10.7554/eLife.28118.025

reduce the response to a variety of cachexigenic challenges (*Steinman and DeBoer, 2013*). This finding provides a potential model system for further studies of the role of MC4R signaling in cardiomyopathy, in which the level of MC4R signaling can be modulated. We are currently testing cardiac function in *Mc4r+/-* mice following doxorubicin treatment.

Homozygous loss of MC4R function, and homozygous loss of proopiomelanocortin gene function, a preprohormone precursor of the endogenous MC4R agonist have both been reported in patients with early onset obesity (*Farooqi et al., 2003*; *Krude et al., 1998*). Based on our findings, these patients, though rare, should be followed for cardiomyopathy. While most of these studies were conducted on *Mc4r−/−* mice, it is important to note that a trend towards reduced contractility was seen in *Mc4r+/-*mice, as is demonstrated for fractional shortening and left ventricular internal dimension (*Figure 1C–E*).

Heterozygous hypomorphic/null alleles of the *Mc4r* are common, appearing in up to 1/1500 individuals (*Wang et al., 2017*). Since most phenotypes characterized in *Mc4r+/-*mice have translated to *Mc4r+/-*patients (*Greenfield et al., 2009*; *van der Klaauw et al., 2016*), patients with heterozygous *Mc4r* mutations should also be followed as our studies suggest they may have an increased risk for

the development of heart failure. Patients with dominant negative *Mc4r* mutations should also be identified and followed as they will more closely reflect the *Mc4r−/−* syndrome (*Tarnow et al., 2008*) (*Biebermann et al., 2003*; *Tao, 2005*). Our finding may thus also have clinical relevance in identifying the etiological factor in certain patients with obesity associated cardiomyopathy. Additionally, our GSEA and doxorubicin treatment findings suggest that common mutations in the MC4R may be an important risk factor for increased sensitivity to cachexia, or possibly even cardiomyopathies induced by cardiotoxic drugs, such as doxorubicin (*Singal and Iliskovic, 1998*).

Additional research will need to be conducted to determine the mechanism by which *Mc4r* deletion causes cardiomyopathy. However, the data shown here provides some important clues. First, since we were unable to observe *Mc4r* expression in adult mouse cardiac tissue, MC4R is likely acting indirectly. One possibility would be an early role for MC4R in cardiomyocyte development; developmentally restricted expression of the *Mc4r* has been observed in E14 through E18 rat heart, for example (*Mountjoy et al., 2003*). Since we observed mitochondrial defects by respirometry in young lean *Mc4r−/−* mice (*Figure 8A–C*) not yet exhibiting significant cardiac dysfunction, it is possible that defective development and/or regulation of cardiomyocyte function contributes to the defective cardiac function seen by 26 weeks of age. Absent a role for early developmental expression in mouse cardiomyocytes, the prominent expression of the *Mc4r* in the central and peripheral nervous system implicates an autonomic mechanism. However, endocrinological mechanisms such as the hyperinsulinemia or abnormal obesity associated growth factor suppression may also need to be considered. Collectively, these studies characterize a novel form of heart failure with clinical importance and raise the need for further examination of how *Mc4r* deletion affects myocardial function in humans.

## Materials and methods

### Mouse handling

All mouse experiments with approval from the Vanderbilt animal care and use committee. All mouse lines were maintained on a C57B6/J background and bred using a het-het mating strategy. Mice were maintained on a 12 hr light-dark cycle and housed at 25°C. Unless other wise noted, mice were fed a chow diet (Lab Diet; St. Louis, MO; S-5LOD - 13.5 kcal% fat, 32.98 kcal% Protein, 56.7 kcal% Carbohydrate) following weaning at 3 weeks of age. For high fat diet studies (Research Diets; New Brunswick, NJ; D12492 - 60 kcal% fat, 20 kcal% Protein, 20 kcal% Carbohydrate), food was administered starting at 4 weeks of age and continued throughout the study. For all post mortem studies, mice were deeply anesthetized with 5 mg/kg tribromoethanol and then sacrificed by decapitation. Tissues were rapidly collected and snap frozen in liquid nitrogen.

### Mouse line information

All mouse lines were maintained on a C57BL/6J background with yearly backcrosses to wild type C57 mice (Jackson Laboratory; Sacramento, California - Jax Stock No: 000664).

*Mc4r−/−* mice (For Details see: Huszar et al. 1997[19]) loxTB *Mc4r* mice (Jax Stock No: 006414)
*Mc4r*-tau-Sapphire (Jax Stock No: 008323)

### Echocardiography

Echocardiography was performed using the VEVO®2100 digital ultrasound system (Visual Sonics; Toronto, Ontario). Studies were performed using the MS400 18–38 MHz transducer. Mice were placed on a heated platform in the supine position and given oxygen throughout the procedure. Fur was then shaved and the transducer oriented to obtain a parasternal long axis image in Brightness-Mode (B-Mode). After obtaining this image, the transducer was rotated to obtain a short axis image at the level of the mid-papillary muscle. Simultaneous one lead ECGs were preformed on the heated platform. Once in position, Motion-Mode (M-Mode) images were taken for further processing. M-mode images were then processed as previously described using the Visual Sonics Software ver2.2. LVIDd, %FS and HR measurements were made in a blinded manner using the LV trace function while LVPWd was obtained using the ruler function.

## High resolution respirometry

After sacrifice, samples were rapidly placed in ice-cold MiR05 buffer (0.5 mM EGTA, 3 mM MgCl$_2$ 6 H$_2$O, 60 mM Lactobionic acid, 20 mM Taurine, 10 mM KH$_2$PO$_4$, 20 mM HEPES, 110 mM D-Sucrose, 1 g/L Essentially Fatty Acid Free BSA) with an additional 2 mM EGTA to chelate extracellular calcium. Samples were permeabilized as previously described(*Talati et al., 2016*). Briefly, samples were separated into fiber bundles using forceps under a dissecting scope. Samples were then moved into a MiR05 +2 mM EGTA +50 µg/mL saponin solution and incubated for 30 min on ice. Samples were then washed twice by incubation in MiR05 buffer with 2 mM EGTA for 15 min. Samples were then blotted dry and weighed so that a 2–4 mg sample was obtained. This sample was then placed in a pre-equilibrated chamber of an O2k Oxygraph (Oroboros Instruments; Innsbruck, Austria) that contained MiR05 buffer without additional 2 mM EGTA. Chambers were then closed and O$_2$ consumption rate was measured in response to sequential additions of 10 mM glutamate and 2 mM malate, 0.05 mM Palmitoylcarnitine, 2.5 mM ADP, 0.25 mM Octanoylcarnitine, 10 mM Succinate and 500 nM CCCP.

## Transmission Electron Microscopy

Samples were fixed with 2.5% glutaraldehyde in a 0.1M sodium cacodylate buffer. Following fixation, the samples were washed in 0.1M sodium cacodylate buffer. After washing, the samples were post-fixed with 1% osmium tetroxide in 0.1M cacodylate buffer and then further washed in a 0.1M cacodylate buffer. The samples were then dehydrated through a graded series of ethanols and infiltrated with epoxy resin. The samples were oriented and the epoxy resin cured in flat-embed molds.

Thick sections of the embedded tissue were cut, stained with 1% Toluidine blue, and reviewed to determine location. When the correct structure was identified in thick sections, a region of interest was selected for thin sections. Thin sections were stained with 2% aqueous uranyl acetate and Reynolds' lead citrate. Samples were viewed using a FEI Tencai T-12 electron microscope (FEI Tencai; Hillsboro, Oregon) operating at 100 keV.

Mouse genotypes were blinded during imaging to avoid sampling bias. Myofiber images were taken from three random EM grids per sample. A trained pathologist subsequently analyzed images in a blinded manner.

## RNA isolation

RNA was isolated using TRIzol$^{®}$ (Thermo Fisher Scientific; Waltham, Massachusetts). 1 mL of Trizol was added to 10 mg tissue samples from the posterior wall of the left ventricle. Samples were homogenized using the TissueLyser II (Qiagen; Venlo, Netherlands) at max speed for 3 min. Samples were then centrifuged for 10 min at 4°C at 10,000 g to eliminate debris. After a 5 min incubation at room temperature, 200 µL of chloroform was added and the samples were shaken vigorously. After an additional 3 min incubation at room temperature, samples were centrifuged at 4°C and 12,000 g to separate phase layers. The upper phase was then mixed with 100% EtOH and added directly onto RNeasy (Qiagen) columns. Columns were used as instructed by the manufacturer and included an on column DNase (Quiagen).

## RNA-seq

Total RNA quality was assessed using the 2100 Bioanalyzer (Agilent; Santa Clara, California). 200 ng of DNase-treated total RNA with a RNA integrity number greater than seven was used to generate polyA-enriched mRNA libraries using KAPA Stranded mRNA sample kits with indexed adaptors (Roche; Basel, Switzerland). Library quality was assessed using the 2100 Bioanalyzer (Agilent) and libraries were quantitated using KAPA Library Quantification Kits (Roche). Pooled libraries were subjected to 75 bp paired-end sequencing according to the manufacturer's protocol (HiSeq3000; Illumina; San Diego, California). Bcl2fastq2 Conversion Software (Illumina; San Diego, California) was used to generate de-multiplexed Fastq files. Read quality was checked using FastQCv0.11.5 (Babraham Institute, Cambridge, UK). Fastq data files were then imported into Galaxy(*Afgan et al., 2016*) and converted to a fastsanger file. A quality control was run and reads were then mapped to the mm10 mouse reference genome with *Tophat* v2.1.0(*Trapnell et al., 2012*). Reads were then assembled into transcripts using *Cufflinks* v2.2.1 and merged using *Cuffmerge* v2.2.1. RMPK values were then quantified using *CuffQuant* v2.2.1. *Cuffnorm* v2.2.1.1 was then used to normalize counts for

transcript length. Differential expression of transcripts was determined using the *Cuffdiff* v2.2.1.3 program and filtered for significance q > 0.05. Once differential gene expression was determined, Gene Set Enrichment Analysis (GSEA; Broad Institute; Cambridge, Massachusetts) was used for pathway analysis and to determine enriched gene sets as previously described(*Subramanian et al., 2005*) (*Mootha et al., 2003*). Pathway analysis was conducted using PANTHER Pathway analysis on significantly different transcripts identified by RNA-seq (*Mi and Thomas, 2009*).

## qRT-PCR

cDNA was generated with iScript™ (BioRad; Hercules, California) according to the manufacturer's instructions. qRT-PCR was performed using POWER Syber master mix (Thermo Fisher Scientific) with the following primers:

| | |
|---|---|
| *mMc4r* F | Cccggacggaggatgctat |
| *mMc4r* R | TCGCCACGATCACTAGAATGT |
| *mPpargc1a* F | GCCGTGACCACTGACAACGAGGC |
| *mPpargc1a* R | GCCTCCTGAGGGGGGAGGGGTGC |
| *mMyl4* F | CGGACTCCAACGGGAGAGAT |
| *mMyl4* R | GCTCCTTGTTGCGGGAGAT |
| *mH19* F | GTACCCACCTGTCGTCC |
| *mH19* R | GTCCACGAGACCAATGACTG |
| *mCytb* F | GTCCACGAGACCAATGACTG |
| *mCytb* R | ACTGAGAAGCCCCCTCAAAT |

All PCR was performed on using standard cycling conditions on a Quantstudio 12 k Flex Real Time PCR system (Thermo Fisher Scientific). Data was analyzed using the ΔΔCT method. All statistics were performed by comparing ΔCT values between groups and plotted as Fold Change ± SEM ($2^{\Delta\Delta CT}$).

## Western blotting

Tissue samples were snap frozen in Liquid $N_2$ and then transferred to −80℃. Tissue was lysed in Radio Immune Assay Buffer (50 mM Tris, 150 mM NaCl, 1% Triton X-100, 0.5% Na-Deoxycholate, 0.1% SDS, cOmplete™ EDTA-Free Protease Inhibitor (Roche) Phos Stop (Roche) using a tissue homogenizer (Pro-Scientific 200; Oxford, Connecticut). Following lysis, protein samples were kept on ice for 30 min, spun at max speed for 30 min and then quantified by Bradford assay (BioRad; Hercules, California). Sample protein content was adjusted to 2 mg/mL and then diluted with 2x Laemmli buffer with 2% BME (1% final). Sampled were then run on a Mini-Protean gel, transferred to PVDF membranes, blocked for 1 hr in TBS-T 5%BSA solution and incubated with respective antibodies overnight (Total OX-Phos Rodent WB Cocktail, 1:1000; Abcam; Cambridge, United Kingdom) (GAPDH, 1:5000; Cell Signaling; Boston, Massachusetts) (GFP, 1:1000; Abcam). After washes in TBST, membranes were incubated with $2^O$ antibodies conjugated (1:10,000; Promega; Madison, Wisconsin) with HRP for one hour. After this incubation, membranes were again washed with TBST, exposed to ECL and imaged (ChemiDoc MP; BioRad; Hercules, California).

## Metabolic studies

Glucose tolerance testing was performed as previously described. 1 week prior to GTT, body composition was obtained. Mice were then habituated to handling for three consecutive sessions. Following habituation, mice were fasted for 4 hr from 2pm-6pm. A basal glucose reading was obtained and mice were then injected with a 2 mg/kg lean mass dose of glucose in PBS. Glucose readings were then obtained at 15, 30,60, and 120 min following injection. Mouse body composition including lean and fat mass was obtained by NMR (mq10 Minispec; Bruker; Billericia, Massachusetts)

## ROS assays

All ROS assays were conducted according to the manufacturer's directions. For ROS (OxiSelect™In Vitro ROS/RNS Assay Kit; Cell Biolabs, Inc.; San Diego, California) assays were conducted on tissue lysates from frozen tissue samples. Samples were weighted (10 mg/sample), homogenized in PBS and centrifuged (10,000 g, 5 min, 4°C). Following sample preparation, assays were conducted according to the manufacturer's directions. MDA content was obtained using MDA lipid peroxidation assay (Abcam) according to the manufacturer's directions. For 4-HNE, the OxiSelect™ 4-HNE Assay Kit (Cell Biolabs, Inc.) was used. Samples were homogenized in RIPA buffer with inhibitors and analyzed according to the manufacturer's directions.

## ATP assay

Fresh tissue was isolated from posterior left ventricular wall, weighted and placed into a 2N perchloric acid solution, homogenized and then incubated on ice for 30 min. Samples were then centrifuged (13,000 g for 2 min at 4°C) and 100 μL of supernatant was added to 500 μL of assay buffer. Samples were then neutralized with 2M KOH, vortexed and centrifuged (13,000 g for 15 min at 4°C) to remove PCA. The resulting supernatant was then used in the Fluorometric ATP Assay Kit (Abcam) according to the manufacturer's instructions.

## Doxorubicin studies

Doxorubicin (Cayman Chemicals #15007) was solubilized in isotonic saline at a concentration of 1 mg/ml. 28 week old *Mc4r+/-*and WT mice received two intraperitoneal injections of 5 mg/kg doxorubicin or vehicle separated by 7 days (cumulative dose of 10 mg/kg). Body weights were measured prior to injection, and at times indicated.

## Statistics

Sample size estimation for echocardiography studies was conducted using the power equation ($\alpha < 0.05$, $\beta = 0.1$, $\Delta\mu = 25\%$ $\sigma = 5$). Remaining sample sizes were estimated based on previous publications. All statistical tests were conducted on the GraphPad Prism 6 software (Scientific Software; La Jolla, California). Data is presented as mean ±standard error of the mean. All data with $p < 0.05$ was considered statistically significant. Statistical nomenclature: * OR ⌢=$p < 0.05$; **$p < 0.01$; ***$p < 0.001$; ****$p < 0.0001$.

## Acknowledgements

We thank Savannah Y. Williams and Heidi Moreno Adams for technical assistance during this research program, and Stephanie King for assistance with the figures. This work was supported by NIH grant F30 DK108476 and T32 GM007347 to MJL, and RO1 HL121174 to JPF.

## Additional information

### Funding

| Funder | Grant reference number | Author |
| --- | --- | --- |
| National Institute of Diabetes and Digestive and Kidney Diseases | F30 DK108476-01 | Michael J Litt |
| National Institutes of Health | T32 GM007347 | Michael J Litt |
| National Heart, Lung, and Blood Institute | RO1 HL121174 | Josh Fessel |

The funders had no role in study design, data collection and interpretation, or the decision to submit the work for publication.

## Author contributions
Michael J Litt, Conceptualization, Data curation, Formal analysis, Funding acquisition, Investigation, Methodology, Writing—original draft, Writing—review and editing; G Donald Okoye, Data curation, Formal analysis, Validation, Investigation, Methodology; Daniel Lark, Data curation, Validation, Investigation, Methodology; Isin Cakir, Mary C Barber, Investigation, Methodology; Christy Moore, Data curation, Validation, Methodology; James Atkinson, Formal analysis, Validation, Investigation; Josh Fessel, Conceptualization, Resources, Data curation, Supervision, Funding acquisition, Investigation, Methodology, Writing—review and editing; Javid Moslehi, Conceptualization, Data curation, Formal analysis, Supervision, Funding acquisition, Investigation, Methodology, Project administration, Writing—review and editing; Roger D Cone, Conceptualization, Resources, Data curation, Formal analysis, Supervision, Funding acquisition, Validation, Investigation, Methodology, Project administration, Writing—review and editing

## Author ORCIDs
G Donald Okoye (iD) http://orcid.org/0000-0003-1078-688X
Roger D Cone (iD) http://orcid.org/0000-0003-3333-5651

## Ethics
Animal experimentation: This study was performed in strict accordance with the recommendations in the Guide for the Care and Use of Laboratory Animals of the National Institutes of Health. All of the animals were handled according to approved institutional animal care and use committee (IACUC) protocol (M/10/358) of Vanderbilt University.

## Decision letter and Author response
Decision letter https://doi.org/10.7554/eLife.28118.027
Author response https://doi.org/10.7554/eLife.28118.028

# Additional files

## Supplementary files
• Transparent reporting form
DOI: https://doi.org/10.7554/eLife.28118.026

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
