## [Decision Letter]

Thank you for submitting your article "Loss of the melanocortin-4 receptor in mice causes dilated cardiomyopathy" for consideration by *eLife*. Your article has been favorably evaluated by Fiona Watt (Senior Editor) and three reviewers, one of whom is a member of our Board of Reviewing Editors. The following individual involved in review of your submission has agreed to reveal his identity: Stephen I. O'Rahilly (Reviewer #2).

The reviewers have discussed the reviews with one another and the Reviewing Editor has drafted this decision to help you prepare a revised submission.

Summary:

Litt and colleagues describe the results from a series of studies investigating the effects on cardiac function in mice lacking melanocortin 4 receptors. The authors used a number of approaches that support the model that lack of melanocortin 4 receptor signaling results in a dilated cardiomyopathy phenotype, which was previously unrecognized in this widely studied model of obesity. They report that, even though MC4R is not expressed in the heart, silencing of its expression provokes a dilated cardiomyopathy. They go on to provide evidence that this myocardial phenotype is both indirect (receptor not expressed in the heart) and independent of the concomitant obesity phenotype. Given the relative frequency of mutations in the MC4R gene in obese subjects, the current results have immediate translational relevance. This is an important observation, which should provoke further studies in humans as it is of potential clinical significance. It raises interesting mechanistic questions which will no doubt be the subject of future research. Thus, the manuscript seems to be a strong candidate and would be of interest to the broad readership of *eLife*. A few suggestions are offered.

Essential revisions:

As the authors well know, a melanocortin obesity syndrome is characterized by a constellation of phenotypes that include hyperinsulinemia and increased linear growth. The increased growth phenotype is evident in the skeletal muscle. Was there also an increase in heart size/weight? If so, does this precede the onset of obesity? Similarly, does the cardiac abnormalities correlate with any changes in insulin sensitivity in the cardiac tissue?

The authors report that, on two different genetic backgrounds, deletion of the MC4R leads to systolic dysfunction, sufficient, on one background at least, to lead to a dilated cardiomyopathy. Given the fact that MC4R deficient human and mice have lower life time blood pressures and might be expected to be relatively protected from cardiac overload, this is a surprising and potentially clinically important finding. The data are clearly presented, the experimental findings look robust and the statistical analysis appears appropriate. The authors demonstrate that this phenomenon is not cell-autonomous and they suggest a range of systemic features of the MC4R deficient state that might contribute (low sympathetic tone, hyperinsulinemia, unsuppressed growth hormone). However they do not "close the loop" in terms of manipulating one or more of these parameters and correcting the cardiac metabolic defect. However, we appreciate that this would involve a very demanding and time-consuming set of experiments.

Why is it that humans with loss-of-function mutations in this gene do not develop systolic heart failure, whereas mice do? To what extent have patients harboring this mutation been phenotyped in terms of cardiac structure and function?

In general, assessment of myocyte hypertrophy will be a required step in the evaluation of ventricular remodeling. This is not done here. Further, given the reported, albeit indirect, evidence of myocyte dropout, it will be critical to evaluate cell (myocyte) hypertrophy as cross-sectional area at multiples points.

It would be interesting to evaluate sympathetic tone in these animals (microneurography, or indirectly by assessing circulating catecholamine levels).

Is the bradyarrhythmia sinus bradycardia? It appears to me that p waves are dropped.

Is the upward increment in oxygen consumption consistent with the increase in ROS production? The authors argue that these processes are both linked and central to myocardial pathogenesis.

---

## [Author Response]

Essential revisions:As the authors well know, a melanocortin obesity syndrome is characterized by a constellation of phenotypes that include hyperinsulinemia and increased linear growth. The increased growth phenotype is evident in the skeletal muscle. Was there also an increase in heart size/weight? If so, does this precede the onset of obesity? Similarly, does the cardiac abnormalities correlate with any changes in insulin sensitivity in the cardiac tissue?

The MC4R obesity syndrome is characterized by an early onset euglycemic hyperinsulinemia and increased linear growth in both mice (Fan. W et al., 2000) and humans (Martinelli, C. et al., 2011). Hyperinsulinemia is associated with myocyte hypertrophy and can lead to heart failure through a variety of mechanisms including oxidative stress, mitochondrial dysfunction, ER stress and impaired Ca^2+^ handling (Boudina S and Abel E. 2007, Guanghong J et al., 2016). In our studies, we find that the heart weight of MC4R-/- mice is no different than lean controls at 6 and 12 weeks of age despite this early onset hyperinsulinemia. We do however see an increase in heart weight at 30 weeks of age, which is following the development of systolic dysfunction. Figure 6 has been updated to include these data. As shown in Figure 6, this increase in heart weight is proportional to an increase in total lean mass and is consistent with appropriate allometric organ scaling (Dawson et al., 2014). However, as shown in Figure 6, the increase in heart weight in MC4R-/- mice is out of proportion with their increased adiposity. MC4R-/- mice have increased heart weight when compared to weight matched DIO animals. This again is in line with the generalized increase in lean mass described in Figure 3. Based on the reviewers’ suggestion and the importance of insulin sensitivity in normal heart function, we have gone on to characterize the insulin sensitivity of MC4R-/- using a phospho-specific pAKT-T308 antibody (Figure 5). In these experiments, no difference in myocardial insulin sensitivity was observed between MC4R-/- and DIO mice. This is consistent with clinical studies that find no difference in insulin studies between MC4R deficient subjects and weight matched controls (Greenfield J et al., 2009). These findings indicate that MC4R-/- mice develop an increased cardiac mass by 30 weeks of age that correlates with an increase in lean mass but not insulin resistance. Based on our echocardiography studies (Figure 1 and Figure 5), this increase in heart mass is not due to concentric hypertrophy – a compensatory response to increased afterload (Hill J & Olson E. 2008), Rather, MC4R-/- mice display eccentric hypertrophy which is more consistent with elevated preload.

The authors report that, on two different genetic backgrounds, deletion of the MC4R leads to systolic dysfunction, sufficient, on one background at least, to lead to a dilated cardiomyopathy. Given the fact that MC4R deficient human and mice have lower life time blood pressures and might be expected to be relatively protected from cardiac overload, this is a surprising and potentially clinically important finding. The data are clearly presented, the experimental findings look robust and the statistical analysis appears appropriate. The authors demonstrate that this phenomenon is not cell-autonomous and they suggest a range of systemic features of the MC4R deficient state that might contribute (low sympathetic tone, hyperinsulinemia, unsuppressed growth hormone). However they do not "close the loop" in terms of manipulating one or more of these parameters and correcting the cardiac metabolic defect. However, we appreciate that this would involve a very demanding and time-consuming set of experiments.

We thank the reviewer for these comments. We agree that the findings of cardiomyopathy in MC4R -/- mice necessitate further investigation, which include a better understanding of the mechanism of cardiomyopathy. For instance, does hypothalamic or autonomic MC4R deletion precipitate cardiomyopathy? Can postnatal MC4R deletion or re-expression induce or rescue cardiomyopathy? These studies necessitate the generation of new animal models and will be the subject of future studies. For this manuscript, we specifically wanted to make sure that the cardiomyopathy phenotype was robust and as a result of MC4R deletion and not due to other confounders (e.g. obesity).

Why is it that humans with loss-of-function mutations in this gene do not develop systolic heart failure, whereas mice do? To what extent have patients harboring this mutation been phenotyped in terms of cardiac structure and function?

To our knowledge, no study has formally examined cardiac function in patients with MC4R LOF mutations. It is important to keep in mind that most MC4R LOF patients have been identified in young children with early onset obesity; just as our phenotype appears in older mice, systolic heart failure may only develop in older MC4R LOF individuals, and may also exhibit variable penetrance. Determining this will require non-invasive and invasive cardiac studies such as cardiac imaging and cardiac catheterization in these patients as they age. Anecdotal reports of acute respiratory distress as a cause of death in MC4R LOF patients were mentioned to us (personal communication with Sadaf Farooqi); however, it is unclear if this is due to cardiomyopathy or some other process (e.g., pulmonary disease). Heterozygosity of null and hypomorphic MC4R mutations has a high prevalence (~1/1500), and future studies will examine differences in cardiac function between MC4R+/- mice and wild type control mice. In experiments added to this revised MS, we show that MC4R+/- mice have increased sensitivity to a sub-cardiotoxic dose of doxorubicin, compared to wild type mice, as measured by weight loss. Preliminary data from one large cohort also indicate increased doxorubicin cardiotoxicity in MC4R+/- vs. WT mice, as well, but this will be the focus of future studies, and a possible Research Advance submission to *eLife*.

In general, assessment of myocyte hypertrophy will be a required step in the evaluation of ventricular remodeling. This is not done here. Further, given the reported, albeit indirect, evidence of myocyte dropout, it will be critical to evaluate cell (myocyte) hypertrophy as cross-sectional area at multiples points.

The reviewers raise an excellent point. Myocardial hypertrophy can be one means by which the heart is capable of responding to various stimuli (including pathological ones) (Hill J and Olson E. 2008). We have examined cardiac hypertrophy by different means. First, on echocardiography, we examined posterior wall thickness and observed no difference between genotypes (Figure 1, and 5O) Second, we assessed cardiac weight of the mouse (Figure 6). Third, based on the reviewer’s excellent suggestion, we characterized myocyte cross sectional area on multiple cross-sections and also found no difference in cross sectional myocyte area. These data are plotted in (Figure 6).

It would be interesting to evaluate sympathetic tone in these animals (microneurography, or indirectly by assessing circulating catecholamine levels).

Reduced 24 hour urine norepinephrine levels, a reliable indicator of serum catecholamines, has been observed in humans (Greenfield J et al., 2009) with MC4R deficiency. Previous studies in rats have also shown that central administration of an MC4R agonist causes an increase in blood pressure that can be blocked with β_1_AR blockers(Kuo, Silva, and Hall, 2003). Studies on MC4R-/- mice have shown they are protected from obesity associated hypertension (da Silva, A et al., 2014) which is driven by increased sympathetic tone (Hall J et al., 2010). Therefore, there is ample evidence in the literature that reduced MC4R signaling results in reduced sympatric tone.

Is the bradyarrhythmia sinus bradycardia? It appears to me that p waves are dropped.

We carefully examined ECG traces of chow fed WT, MC4R+/- and MC4R-/- mice. In the traces presented in Figure 2, the p waves are indeed dropped in MC4R-/- mice. This finding must be tempered by the fact that this is a single external lead taken during echocardiograms. Therefore, it is difficult to safely conclude whether or not the bradycardia is sinus in nature. Future studies have been planned to examine this aspect of MC4R-/- cardiac function using invasive electrophysiological recording techniques.

Is the upward increment in oxygen consumption consistent with the increase in ROS production? The authors argue that these processes are both linked and central to myocardial pathogenesis.

Reactive oxygen species production in the heart is byproduct of mitochondrial respiratory activity by complex I and complex III which can result in heart failure (Brown D. et al., 2017). We note a global increase in ROS production that is proportional to the increase in respiratory capacity and increases with age. Older MC4R-/- mice have a 2 fold increase in tissue ROS (Figure 10) and a 2 fold increased maximal respiratory activity (Figure 7). Younger MC4R-/- mice have a 1.5 fold increased in ROS (Figure 10) and a 1.5 fold increase in maximal respiratory activity (Figure 8). The link between these two findings remains untested. However, both ROS formation and measured state 3 respiratory activity go up in response to increased mitochondrial membrane potential (Nicholls D, 2004) and contributes to age associated tissue damage. While an increase in mitochondrial membrane potential is a possible source of ROS production in these animals, this hypothesis remains to be tested.